# Float Current Analysis for Fast Calendar Aging Assessment of 18650 Li(NiCoAl)O2/Graphite Cells

**Michael Theiler, Christian Endisch and Meinert Lewerenz ***

Research Group Electromobility and Learning Systems, Technische Hochschule Ingolstadt, D-85049 Ingolstadt, Germany; Michael.theiler@thi.de (M.T.); Christian.Endisch@thi.de (C.E.)
* Correspondence: Meinert.Lewerenz@thi.de

**Abstract:** Float currents are steady-state self-discharge currents after a transient phase—caused by anode overhang, polarization, etc.—is accomplished. The float current is measured in this study with a standard test bench for five 18650 cells (Samsung 25R) at potentiostatic conditions while the temperature is changed in 5 K steps from 5 °C to 60 °C. The entire test is performed in about 100 days resulting in 12 measurement points per cell potential for an Arrhenius representation. The float current follows the Arrhenius law with an activation energy of about 60 kJ/mol. The capacity loss measured at reference condition shows a high correlation to the results of float currents analysis. In contrast to classical calendar aging tests, the performed float current analysis enables determining the aging rate with high precision down to at least 10 °C. Returning from higher temperatures to 30 °C reference temperature shows reducing float currents at 30 °C for increasing temperature steps that may originate from an hysteresis effect that has to be investigated in future publications.

**Keywords:** 18650; NCA; graphite; float current; self-discharge; calendar aging; Arrhenius

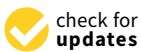



## 1. Introduction

Lithium-ion batteries are nowadays part of most storage applications such as e-mobility, grid service or even power tools or mobiles. Especially providers offering applications with expected lifetimes ranging from 10 to 20 years have a high demand for long lasting cells. Mostly, these cells, used, e.g., for e-mobility, are in operation only a short period and rest for most of the time in, e.g., Europe at rather low temperatures. Thus, the aging rate is mostly low and is therefore a challenge to be predicted via capacity tests, where the values are superimposed with reversible effects caused by anode overhang [1–3], polarization [4] and increased internal resistance [5,6]. Thus, usual tests are not performed for temperatures below 20 °C or the results are hard to be interpreted [2,5–9]. Exceptions are found in the works of Naumann et al. [10], Schmitt et al. [11] and Wu et al. [12] where tests down to 0 °C are performed with superimposed significant influence of the anode overhang and the temperature during a check-up. The outcome is very questionable, as most of the influence will be originated from anode overhang, the heating up during reference tests at room temperature or by the check-up itself.

Additionally, higher temperatures, occurring during operation and/or hot days, induce significant aging of the cells. Therefore, a dependence of the aging rate on the temperature and the cell potential needs to be assessed with a fine resolution, as reported by Rumberg et al. [13], Hoog et al. [14] or Bouchima et al. [15] in 3D representations (T vs. SOC/V vs. capacity losses). It is apparent that the effort for classical check-up tests is high in terms of necessary test cells, test channels, temperature chambers and test duration.

An alternative approach to evaluate faster the aging rate at various conditions is the high precision coulometry (HPC) [16,17]. Using this method, however, leads to comparable high capital costs, and the state-of-charge (SOC) is changing throughout the test, so that reversible effects caused by, e.g., the anode overhang [18] have to be considered by

additional models. Therefore, high precision coulometry is more suitable for the estimation of the cyclic aging rather than calendar aging.

A promising candidate for calendar aging tests is the float current analysis. For this method, the steady-state current in a potentiostatic phase is evaluated after polarization and anode overhang effects are balanced out. Moreover, the float current is independent of the internal resistance: On the one hand, the measured currents are very low and with this the overpotential by increase of internal resistance in the order of some μV. On the other hand, the constant voltage charge is limited by the self-discharge occurring [19]. To summarize, the float current is the constant self-discharge after transient-effects are compensated.

A correlation of float currents to capacity loss rate is, up to now, reported only for LiFePO$_4$-graphite cells [19]. In this publication, the float current tests wereperformed at 3.6V (100% SOC). Assuming that capacity loss is caused by solid electrolyte interphase (SEI) formation [20], as is typically the case, intercalated active lithium-ions from the graphite are passivated in the SEI. As the slope of the open circuit potentials of the graphite potential is hardly zero, the existence of float currents cannot be explained directly by passivating lithium on the anode. A potential explanation is given in the publication by a parallel self-discharging shuttle reaction between the anode and the cathode triggered by SEI formation. Deshpande et al. [21] proposed additional side reaction by electrolyte decomposition at the graphite anode leading to a shuttle reaction at room temperature. Self-discharge was also investigated by Zilberman et al. [22] by evaluating open circuit voltages over time that correlate to loss of active lithium in 18650 cells. They associated the self-discharge currents in μA range with coupled side reactions of the anode and cathode and excluded internal shorts as the self-discharge follows the Arrhenius law. At very high cell potentials beyond the safe operational potential limits, the cathodic side reactions are dominating, and CO$_2$ formation was reported by Xiong et al. [23,24] or not-specified parasitic reactions from 4.2 to 4.6 V by Zeng et al. [25].

Theoretically, the overpotential during the potentiostatic float current measurement could trigger a reaction and is not existing during open circuit storage. For SOC of about 75% at 25 °C and about 90% at 60 °C, Deutschen et al. [26] showed self-discharge losses during open circuit storage that stabilize after long measurement. The equivalent float current calculated from the constant loss by self-discharge neglecting the transient part is in the order of 0.5%/100 days and 5.0%/100 days and fits to results reported in the literature [3] for comparable chemistries excluding anode overhang effect. This is in accordance to the findings of Zilberman et al. [22].

In this contribution, we present a fast parameterization of the calendar aging rate as a function of the cell potential and the temperature for five 18650 NCA/graphite cells. The resolution at low temperatures is highlighted within 0–10 °C. Finally, the path dependence of float currents is discussed. The data are correlated for each cell separately by comparing capacity losses rate and float current. The aging and the float current are further discussed by the active material losses obtained from differential voltage analysis. Moreover, the trend temperature dependence over SOC for all five cells is evaluated by means of the Arrhenius law. The cell-to-cell variation during calendar aging is beyond the scope of this publication and is expected to be comparatively low, especially for lower SOCs, as shown in many publications before [3,5,9,27]. The test strategy including a low temperature and a high temperature sequence is presented in Section 2. Then, at first, the check-ups are evaluated with standard methods in Section 3.1, followed by the float current analysis in Section 3.2, Section 3.3, Section 3.4. In Section 3.5, results of both methods are compared with each other. Finally, in Section 3.6, the hysteresis and reproducibility at 30 °C is discussed.

## 2. Materials and Methods

### 2.1. Electrical Characterization and Test Setup

In this study, 5 cylindrical 18650 cells of the type Samsung 25R were investigated while stored in an ATT climate (DY110) chamber using an LBT21084 test system from Arbin (5 A,

0–5 V). For this cell type, a post mortem study is found in the work by Lain et al. [28]. The specifications according to datasheet and their work are listed in Table 1.

**Table 1.** Specifications of the used battery.

| Manufacturer | Samsung |
|---|---|
| Type | INR18650-25R |
| Cathode | $Li(Ni_{0.8}Co_{0.15}Al_{0.05})O_2$ + $Li(Ni_{0.6}Mn_{0.2}Co_{0.2})O_2$ |
| Anode | Graphite + Silicon |
| Charge cut-off voltage | 4.2 V |
| Discharge cut-off voltage | 2.5 V |
| Nom. Capacity @ 0.5C | 2.5 Ah |
| Max. discharge current | 20 A |
| Energy density | 216 Wh/kg |
| Voltage (SOC) before test | 3.52 V (20%) |

Check-ups were performed at 30 °C reference temperature for all cells to compare the classical method with float current technology (see Table 2). In the beginning of the check-up, the cells were charged to 4.2 V until the current is below 0.25 A (CCCV). After a 15 min break, the cells were discharged with 1 C and were recharged after 15 min with the latter charge protocol. The 1 C recharged step was also followed by a CV phase according the first charge step. This was followed by a 0.1 C discharge/charge cycle in constant current mode with a pause of 15 min before discharge and charge. Afterwards, two pulse tests were performed at 4.2 and 3.7 V to determine the internal resistance. The 1 C discharge pulse was executed after a 15 min rest period, and the internal resistance was evaluated from the voltage drop during the first 10 s. The check-up ended with a fully discharge with 0.1 C down to 2.5 V.

**Table 2.** Check-up test.

| Step | Mode | Values | Pause |
|---|---|---|---|
| 1 | CCCV charge | at 1 C to 4.2 V until I < 0.25 A | 15 min |
| 2 | CC discharge | at 1 C to 2.5 V | 15 min |
| 3 | CCCV charge | at 1 C to 4.2 V until I < 0.25 A | 15 min |
| 4 | CC discharge | at 0.1 C to 2.5 V | 15 min |
| 5 | CC charge | at 0.1 C to 4.2 V | 15 min |
| 6 | CC discharge | at 0.1 C to 3.7 V | 15 min |
| 7 | Pulse test | 1 C discharge; 10 s | |
| 8 | CC discharge | at 0.1 C to 2.5 V | |
| 9 | CC charge | at 0.1 C to float voltage | |

After the check-up, the cells were charged at 0.1 C to the corresponding float voltages listed in Table 3 with the corresponding SOCs. After a period of 165 days and even earlier, the equalization process of the active part of the anode and the passive anode overhang was concluded and the steady-state of the float current was reached. Due to problems with the appropriate measurement range, the float currents of the initial 165 days could not be evaluated.

**Table 3.** Test matrix.

|  | Cell Potential | SOC at Begin of Test |
|---|---|---|
| Cell 1 | 4.10 V | 92% |
| Cell 2 | 3.85 V | 63% |
| Cell 3 | 3.70 V | 47% |
| Cell 4 | 3.52 V | 20% |
| Cell 5 | 3.40 V | 9% |

### 2.2. Float Current Measurement

After a check-up, the float current measurements were evaluated from Day 165 on while stepwise changing the temperature. Two temperature sequences were tested, as depicted in Figure 1. The low temperature sequence (blue) was from 5 to 30 °C in 5 K steps to evaluate the lower temperature behavior of the cell, lasted approximately 36 days, and ended with a check-up at 30 °C. Thereafter, the high temperature sequence (red) was from 30 to 60 °C with 5 K steps as well, returning to 30 °C after each step (30, 35, 30, 40, 30, etc.). The reference temperature at 30 °C was used to evaluate changes in absolute values caused by strong aging. Finally, the test finished with a last check-up.

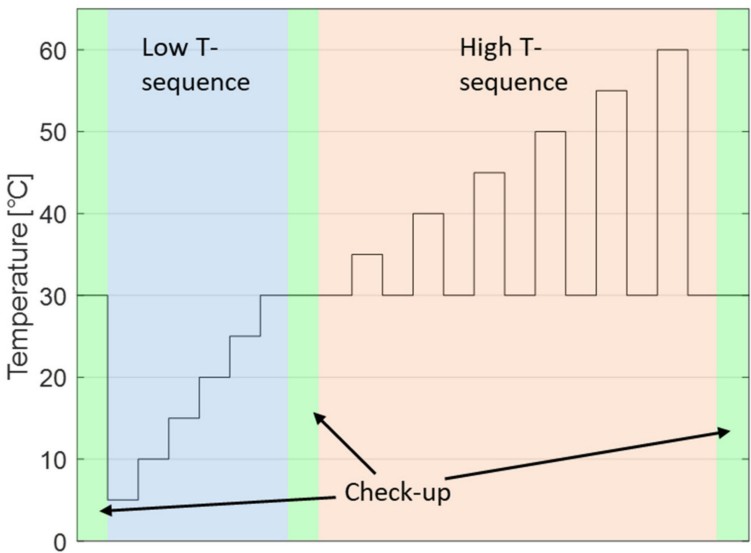

**Figure 1.** Test strategy described in temperature over test steps.

The float current was not evaluated using the direct current since the current jumps in the range of ±1 mA with the highest current measurement resolution of about ±1 μA. Avoiding large datasets and data filtering, the derivative of within the test bench averaged Coulomb counter for charge and discharge is used as float current. Therefore, the internal Coulomb counter isnamed in the following "float capacity".

### 3. Results and Discussion

At first, the test results of the standard check-up routine are evaluated at different aging states by capacity loss, internal resistance change and differential voltage analysis. This is followed by the float current analysis measurement strategy and evaluation. Thereafter, the capacity and the float current test results are compared to each other qualitatively and quantitatively. In the final section, the absolute values at 30 °C in dependence of the previous temperature is presented.

*3.1. Check-Up Results*

3.1.1. Capacity Loss and Pulse Resistance

The capacity loss over the time evaluated from the standard check-up procedure for all cells is depicted in Figure 2a. The initial phase until 165 days is slightly distorted due to problems with the test bench, and repeated and uncompleted check-ups are neglected. However, in the first 50 days, a clear effect from the anode overhang is visible. Potentials during tests close to the delivery voltage 3.52 V follow a rather linear trend in the beginning as no potential difference in the anode causes lithium-ion transport from or to the anode overhang. With increasing test potentials, there is an increasing initial drop before the linear phase starts. After 165 days, the anode overhang effect is concluded and the irreversible capacity loss, determined by the slope, increases with cell potential and is higher in the high temperature sequence (red) compared to the low temperature sequence (blue).

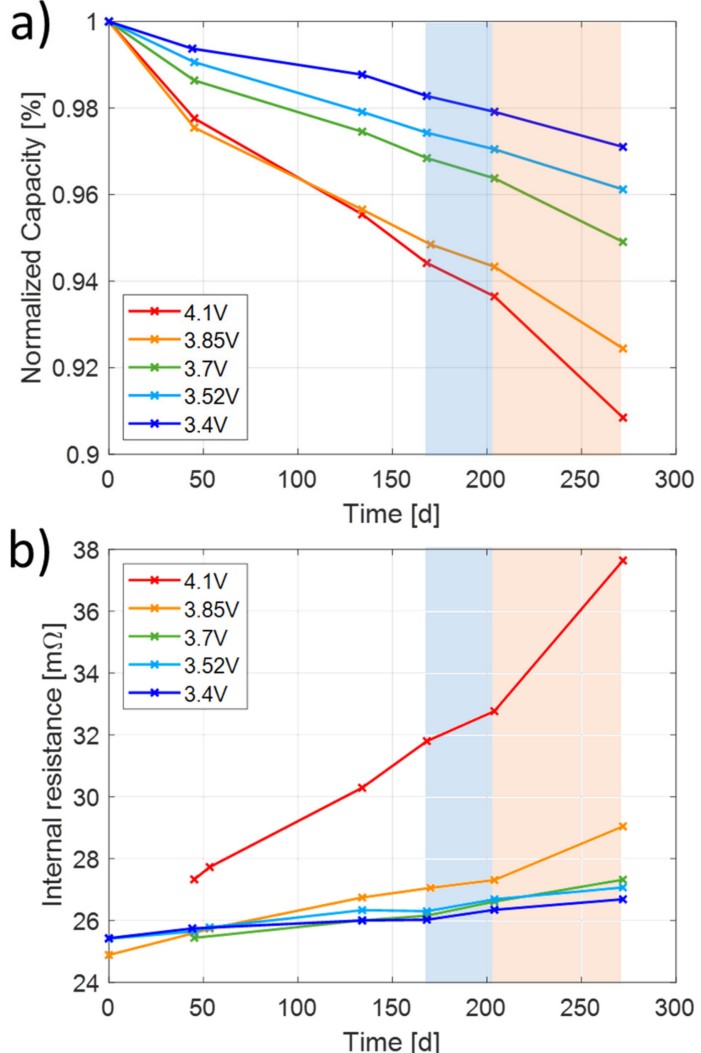

**Figure 2.** Normalized (**a**) capacity tests and (**b**) pulse tests (evaluated at 3.7 V after 10 s) over test time for all five cell potentials. The low temperature cycle is in blue and the high temperature cycle in red.

Figure 2b shows the development of the internal resistance over the time, evaluated from the pulse tests measured at 3.7 V. The results are comparable to the pulse tests performed at 4.2 V (not shown here). The internal resistance rises over calendar aging for all cells. While the resistance rise rate remains similar for the cells, kept at voltages up to 3.7 V, a small slope increase for the cell at 3.85 V and a significantly higher slope increase

for 4.1 V cell are apparent. Unfortunately, in the first check-up for 4.1 V, the pulse test was erroneous and could not be evaluated.

As shown by Baghdadi et al. [29] for $LiNi_{0.8}Co_{0.15}Al_{0.05}O_2$/graphite, the capacity loss and the resistance increase more strongly with higher cell potentials and higher temperatures. This impact is most severe at higher potentials.

### 3.1.2. Differential Voltage Analysis (DVA)

For the electrode specific characterization of the aging, the differential voltage analysis of a charging sequence at 0.1 C is performed analogously to that in [30,31]. The DVA pattern of the studied cell shows four characteristic features (Figure 3): two belong to the cathode and two to the anode. Fitting the distances between these characteristics relative to the initial curve, the percentage losses of anode and cathode active material and slippage between anode and cathode can be calculated. These values are obtained using a self-written Matlab-model by fitting shifted and stretched half-cell curves of the anode and the cathode to the full cell curve. The degradation caused by loss of active lithium (loss of lithium inventory LLI), quantified by slippage between anode and cathode curves, is given in Figure 4a. Figure 4b shows the losses assigned to the anode (LAM-Anode), while Figure 4c shows the loss assigned to the cathode (LAM-Cathode). The trend of LLI is comparable to capacity loss in Figure 2a. In the low temperature sequence, the small increase in Figure 4a,b is below the resolution of the DVA evaluation method, so that it can be considered that there is no significant loss of active material on the anode and cathode. During the high temperature sequence, mainly the cathode active material is aging, while the anode active material is hardly aging. The loss of cathode active material increases with cell potential. Keil et al. [27], who also investigated NCA/graphite, reported mainly LLI losses and only small losses of active material at 25–55 °C. Only for SOC above 80% additional aging at the cathode by electrolyte decomposition is found by end-point slippage. For NCA, micro cracks were reported by Lang et al. [32], enhancing SEI formation. In the low temperature sequence, only SEI formation is observed, while, in the high temperature sequence, the cathode degrades significantly, too.

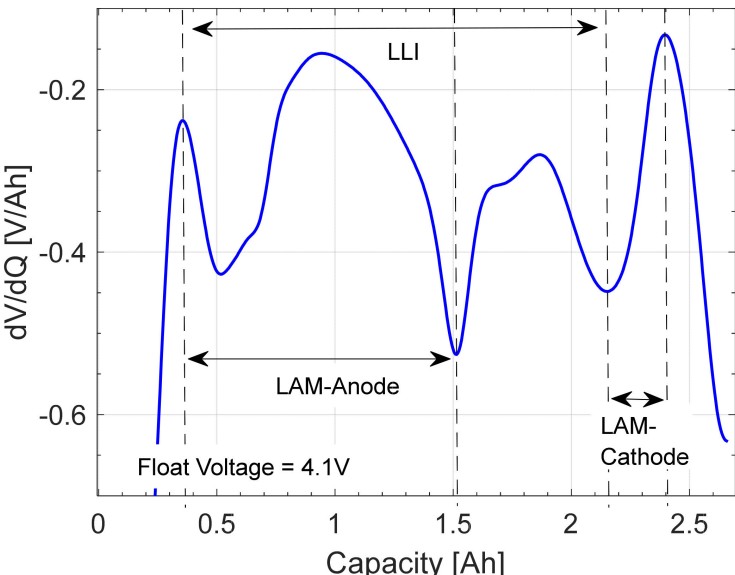

**Figure 3.** Exemplary dV/dQ-curve of the test cells to highlight the characteristic features of the anode and the cathode.

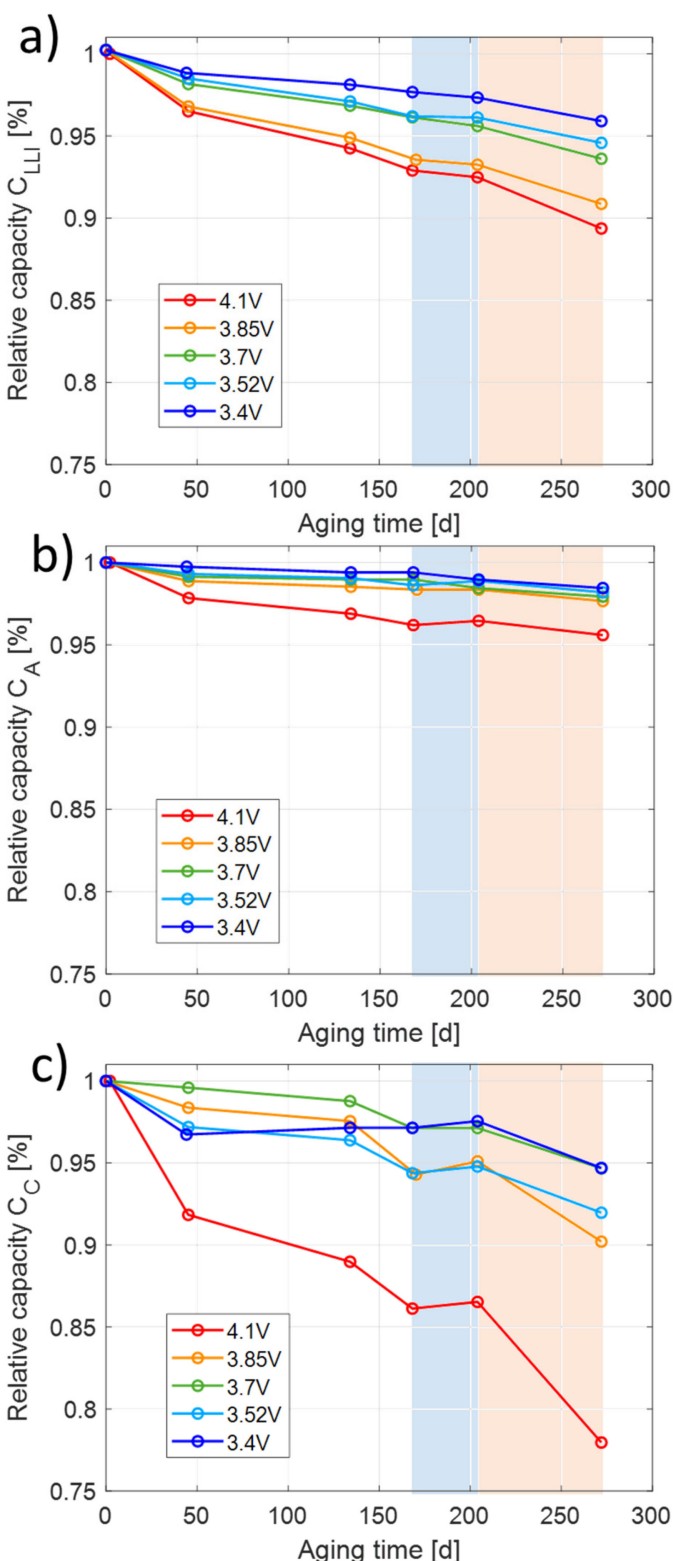

**Figure 4.** Loss of (**a**) active lithium $C_{LLI}$, (**b**) anode active material $C_A$ and (**c**) cathode active material $C_C$ evaluated from DVA over test time for all five cell potentials. The low temperature cycle is in blue and the high temperature cycle in red.

Only LLI losses such as SEI formation are directly associated to loss of extractable capacity. Moderate losses of anode and cathode active materials do not necessarily lead to capacity fade as both electrodes are initially oversized in capacity and as this oversize is further increased by LLI loss over aging. However, if the deactivated active material

encapsulates active lithium, this will lead to capacity fade but will appear as LLI losses and therefore cannot be separated from SEI formation on the anode. Thus, the contribution of loss of cathode active material on capacity fade is limited and not fully determinable by DVA.

## 3.2. Float Currents

The float currents calculated from the derivative of the float capacity are shown in Figure 5a for the low and in Figure 5b for the high temperature sequence. After derivation, the float capacity is filtered with a 2000-sample-wide moving average window. The float current increases with cell potential at a given temperature and with temperature at a given potential, as is the case in Figure 2a for capacity loss. For a more distinct view, the float currents are determined and plotted in the following sections. Between two temperature steps, the influence of the entropy effect [33] becomes visible as the open circuit potential is shifting by some mV, leading to a float current peak. The sign depends on whether the temperature increased or decreased and on the intensity on the entropy coefficient. The evaluation of the float currents follows in the next sections.

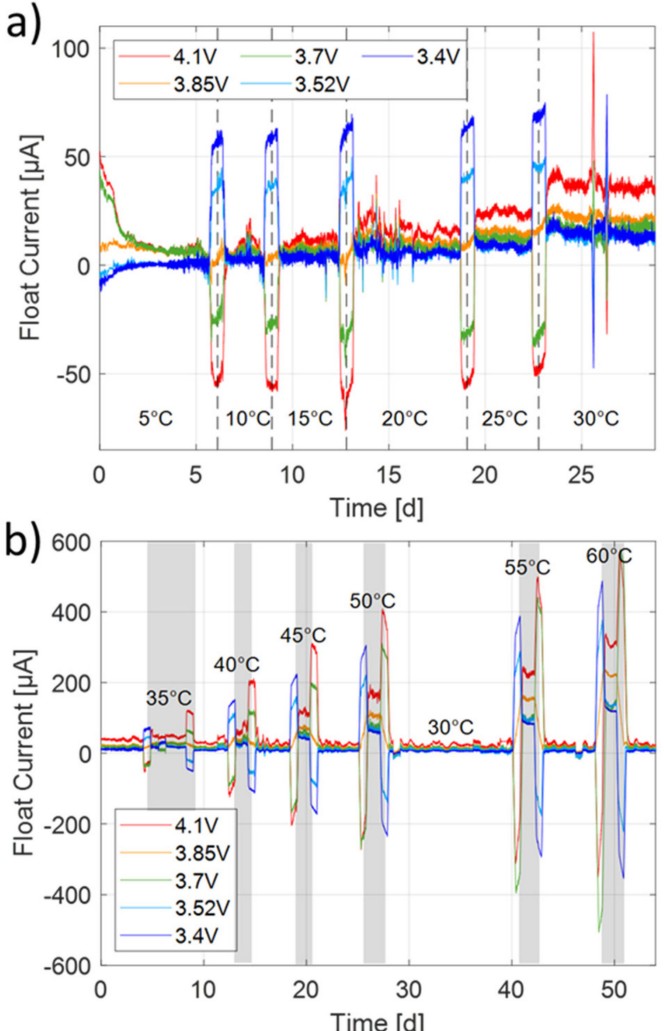

**Figure 5.** Float current over time for five different test voltages for: (**a**) the low temperature cycle from 5 to 30 °C; and (**b**) the high temperature cycle from 30 to 60 °C. In grey bars, the parts higher than 30 °C are highlighted for better understanding. For a better overview, the first few days including the transient part after the check-up are excluded from the graph (7 days in the low temperature sequence and 14 days in the high temperature sequence).

### 3.3. Float Currents Fitting Strategy

For the determination of the float current at a specific temperature, a linear fit is applied to the float capacity after the linear slope has stabilized. The results for the different float potentials of the low temperature sequence are given in Figure 6a–e and for the high temperature sequence in Figure 6f. Exemplarily, Cell 1 aged at 4.1 V in the temperature range from 5 to 30 °C is discussed (Figure 6a). The blue curve represents the measured capacity of the float current. The steady-state float current is superimposed mainly by two measurement artifacts.

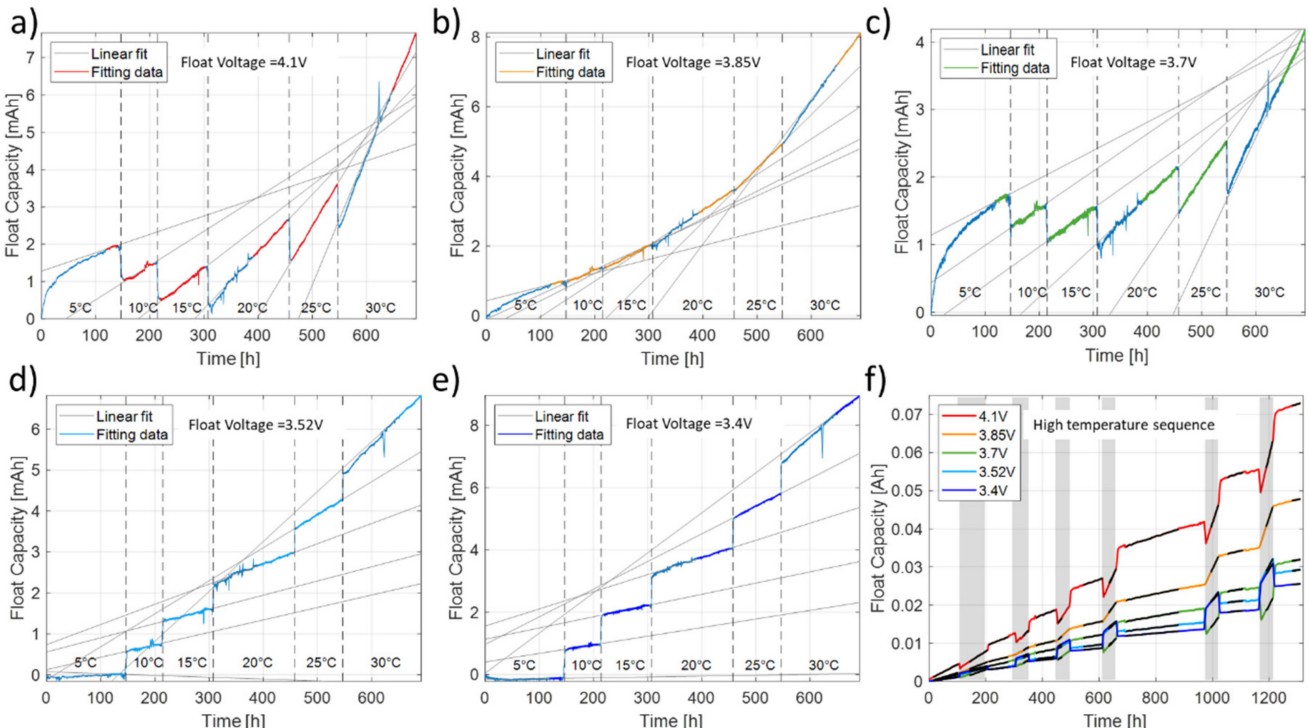

**Figure 6.** Float capacity over time for the low temperature sequence: (**a**) 4.1 V; (**b**) 3.85 V; (**c**) 3.7 V; (**d**) 3.52 V; and (**e**) 3.4 V; and in (**f**) for all cells in the high temperature sequence. Linear fits in black to the float capacity in the specific color to determine the float current at each temperature. In the specific cell colors, the parts are highlighted that are used for the fitting, avoiding transient and noisy parts. The dashed line highlights a temperature change.

On the one side, there are the entropy caused effects at the temperature jumps >5 K leading to temporal higher or lower currents depending on the cell potential. The entropy coefficient is positive for 3.52 Vand 3.4 V, negative for 3.7 Vand 4.1 V and shows hardly any influence for 3.85 V. On the other side, within a temperature step, with a low temperature variation, a coherent fluctuation with the same sign over all tested cells is observed. They are linked to fluctuations in voltage output of the test station correlating to temperature fluctuation in the test container of typical ±2 K. In both cases, the accumulated voltage output is not supposed to change the average output of the float capacity as it is fitted. Therefore, a fit routine is performed over several days after the float capacity has stabilized.

After each temperature step in Figure 6a, the float capacity is reduced due to entropy effects as the open circuit voltage rises. Only in the first step from 30 to 5 °C is the trend inverse and the time to reach a stable current is significantly longer. This is explainable by the higher temperature drop and slow equalization processes at 5 °C due to, e.g., anode overhang effect. The red colored part is used for linear fit. Transient parts and noisy parts as described before are not used for the fitting.

The error of the determined float current is ±1 μA due to precision of the cycler. Additionally, the error of the evaluation method by linear fit is estimated to be an additional ±1 μA. For higher currents, short measurement times and noisy temperatures, the error is

assumed to increase. Only positive float capacities are observed, so that errors cannot lead to negative float currents.

Due to local and time-related temperature variations in the climate chamber of $\pm 1.5$ K, the cell temperature is used instead of the set point temperature of the chamber. For evaluation, at least one day is fitted to exclude night and day temperature changes and the initial part is neglected in the first 10 h or when the float capacity is approaching a linear trend.

The influence of a previous check-up changes the SOC in the anode overhang or the active anode leading to small but long-lasting compensation currents, influencing the float current with superimposed positive currents for higher SOC and negative currents for lower SOC (not shown here). Therefore, the float current after a checkup is evaluated after more than 11 days.

### 3.4. Float Current Evaluation

At first, the dependence of the cell potential over the temperature is evaluated. Figure 7 depicts the entire temperature range from 5 to 60 °C. The float current increases smoothly with higher cell potential and higher cell temperature.

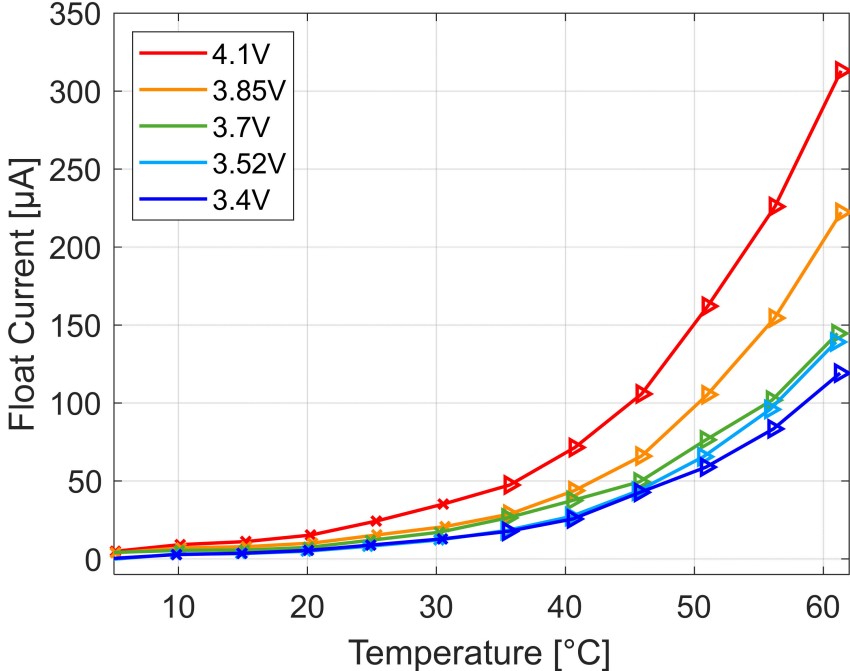

**Figure 7.** Float current over temperature for five different cell potentials with the expected uncertainty by the measurement. The data are fitted with an e-function.

Figure 8 shows only the low temperature behavior of the float current analysis in the range of 0 to 25 °C. Although the results at 5 °C are less trustworthy as the steady-state of the float current was not reached, we can deduce that this method is able to measure capacity fade rates for this 18650 cell down to about 0 °C at potentials greater than 3.7 V. For lower potentials, we assume the limit to be at about 5 °C. For larger cells with higher capacity and active surface, even lower temperatures will be measurable with the accuracy of the here used test bench.

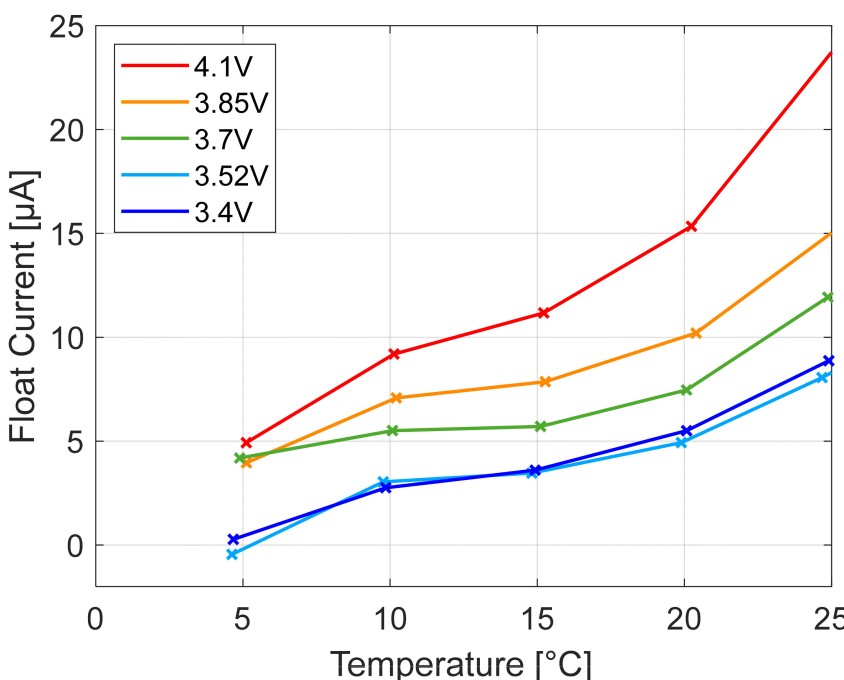

**Figure 8.** Zoomed representation of Figure 7 of the float current over temperature for five different cell potentials.

The relation at a given cell potential in Figure 7 follows an e-function of an electrochemical reaction.

$$I_{float}(\text{T}) \; \sim \; e^{-\frac{E_A}{RT}} \qquad (1)$$

$R$ is the universal gas constant and $E_A$ is the activation energy. The according representation is the Arrhenius plot given in Figure 9 Here, one can observe that at 5 °C the cells were not yet in a steady-state so they appear as a clear outlier. For each measurement point, the method-specific estimated error is added with error bars.

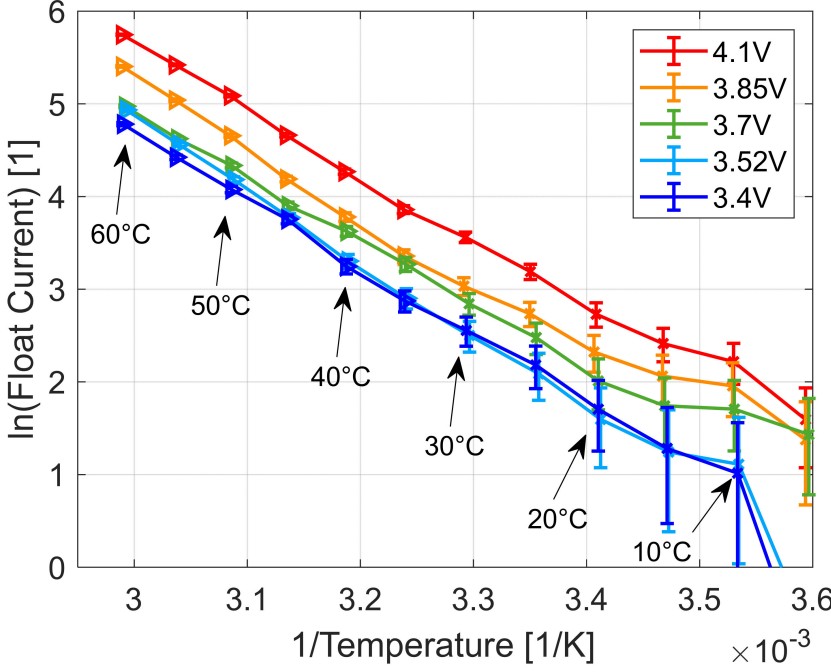

**Figure 9.** Arrhenius representation for all five cell potentials with the expected uncertainty by the measurement.

The activation energy $E_A$ is calculated from the slope of the Arrhenius representation by

$$E_A = -\text{R} \cdot \text{slope} \tag{2}$$

and is shown in Figure 10 over temperature for five voltages. The activation energy is not constant and varies within 30–90 kJ/mol (0.31–0.94 eV) with an average value of ~60 kJ/mol (0.62 eV). In the review by Waldmann et al. [34], the regime for de-solvation and moving lithium-ions into the graphite lattice is measured as 0.3–0.7 eV. This fits the results of an earlier paper performing float current tests by Lewerenz et al. [3] with 64.5 kJ/mol (0.67 eV) for $LiMn_{1/3}Ni_{1/3}Co_{1/3}O_2$/graphite cells. The trends are nearly the same for all test potentials but do not sort according to cell potential. Higher variation is observable at lower temperatures and SOC, but the errors are also higher for these test conditions.

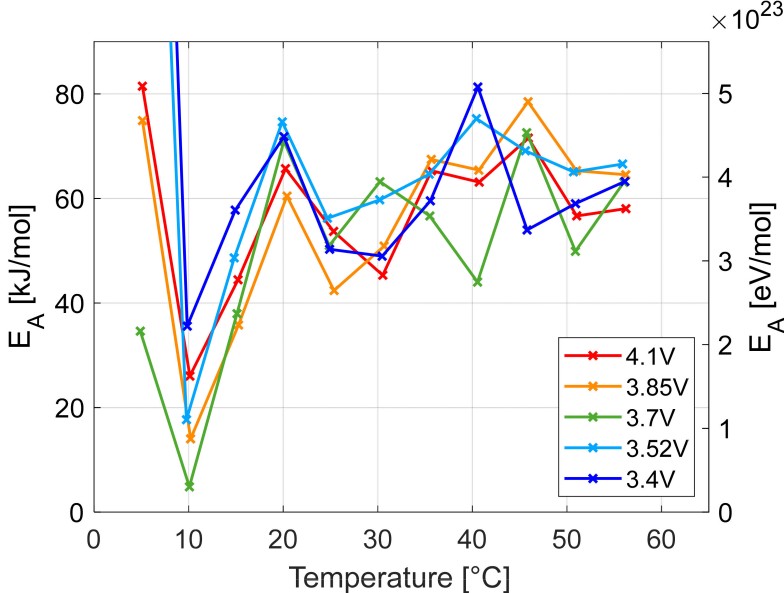

**Figure 10.** Activation energy EA over temperature in the range of 5–60 °C setpoint temperatures.

A representation of the float current over the cell voltage for different setpoint temperatures is given in Figure 11 The float current increases with temperature and with cell voltage but the increase over the cell potential is not smooth but increases in at least two different slopes. The cell potential at the transition between these parts shifts to lower potential with increasing temperature. Therefore, the optimum maximum voltage to reduce strong aging at a given temperature can be determined to guarantee long life.

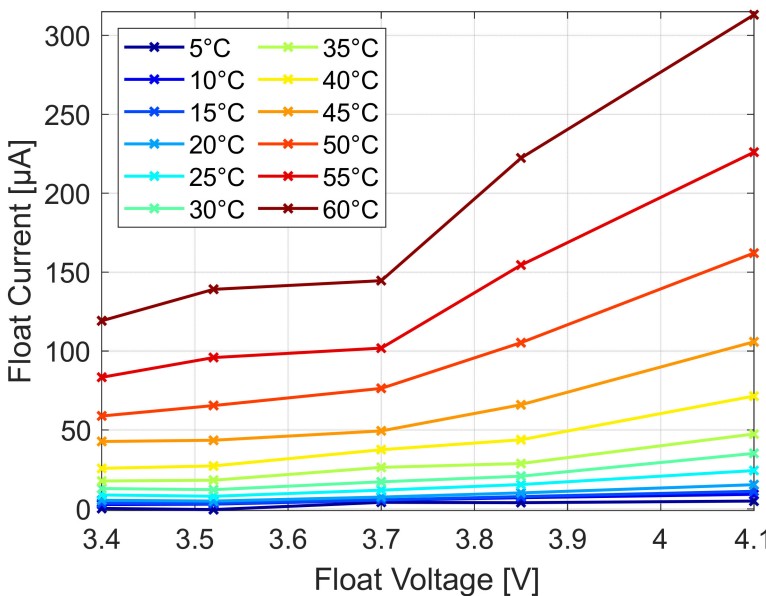

**Figure 11.** Float current over cell potential for several set-point temperature steps in the range of 5 to 60 °C setpoint temperature.

The results in Figures 7 and 11 are merged in a 3D plot in Figure 12. The float current pattern has comparable trend to the expected capacity loss representations for $Li(Ni_xMn_yCo_z)O_2/$ graphite [35] and $Li(Ni_6Mn_2Co_2)O_2/$graphite [13] with the difference that they did not include the anode overhang effect leading to a graphite pattern over cell potential. However, with float currents, the anode overhang effect is excluded. At low temperatures, the aging is independent of the float voltage, while, at high temperatures, even cells with a low potential suffer from strong degradation. As expected, we obtained the highest float currents at high temperatures and high cell potentials.

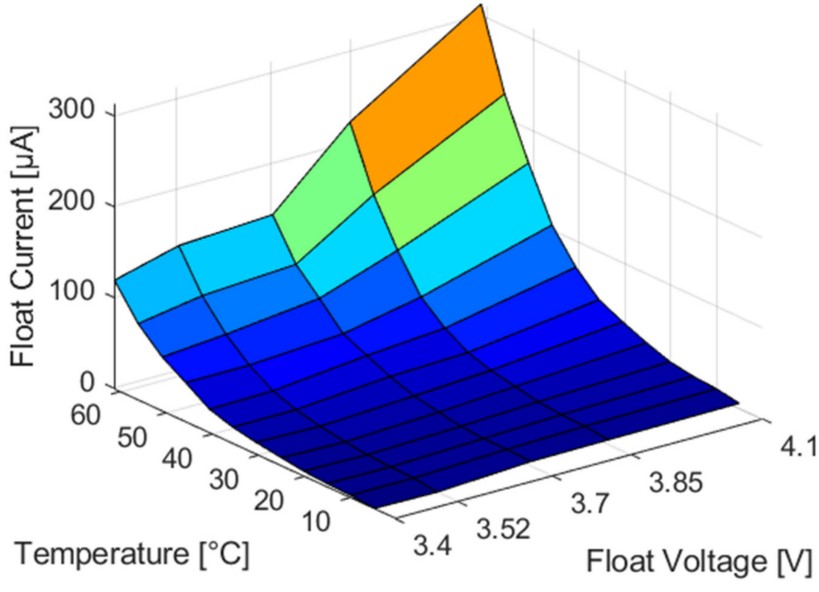

**Figure 12.** 3D plot of the previous figure float current vs. temperature and float voltage.

*3.5. Float Current and Capacity Loss at 30 °C*

At 30 °C reference temperature, a comparison between capacity loss via capacity test and float current is evaluated for the low and high temperature cycles. Therefore, the difference of the extractable capacity $\Delta C_{CU}$ before and after the temperature cycle is

compared to the accumulated capacity during floating $\Delta C_{Float}$ calculated by the sum of the products of current $I_{Float}(t_i)$ and its duration $\Delta t_i$ in each constant temperature phase i.

$$\Delta C_{CU} = C_0 - C_{end} \tag{3}$$

$$\Delta C_{Float} = \int_{t_0}^{t_{end}} I_{Float}(t) \cdot dt = \sum_i \Delta t_i \cdot I_{Float}(t_i) \tag{4}$$

Figure 13 presents the results for the low (blue) and high temperature sequence (red) and the combination of both (black). In Figure 13a the average float current and in Figure 13b the corresponding capacity loss in the period is given. The error of the float current is assumed to be $\pm 2$ µA and for the check-up to be $\pm 2$ mAh.

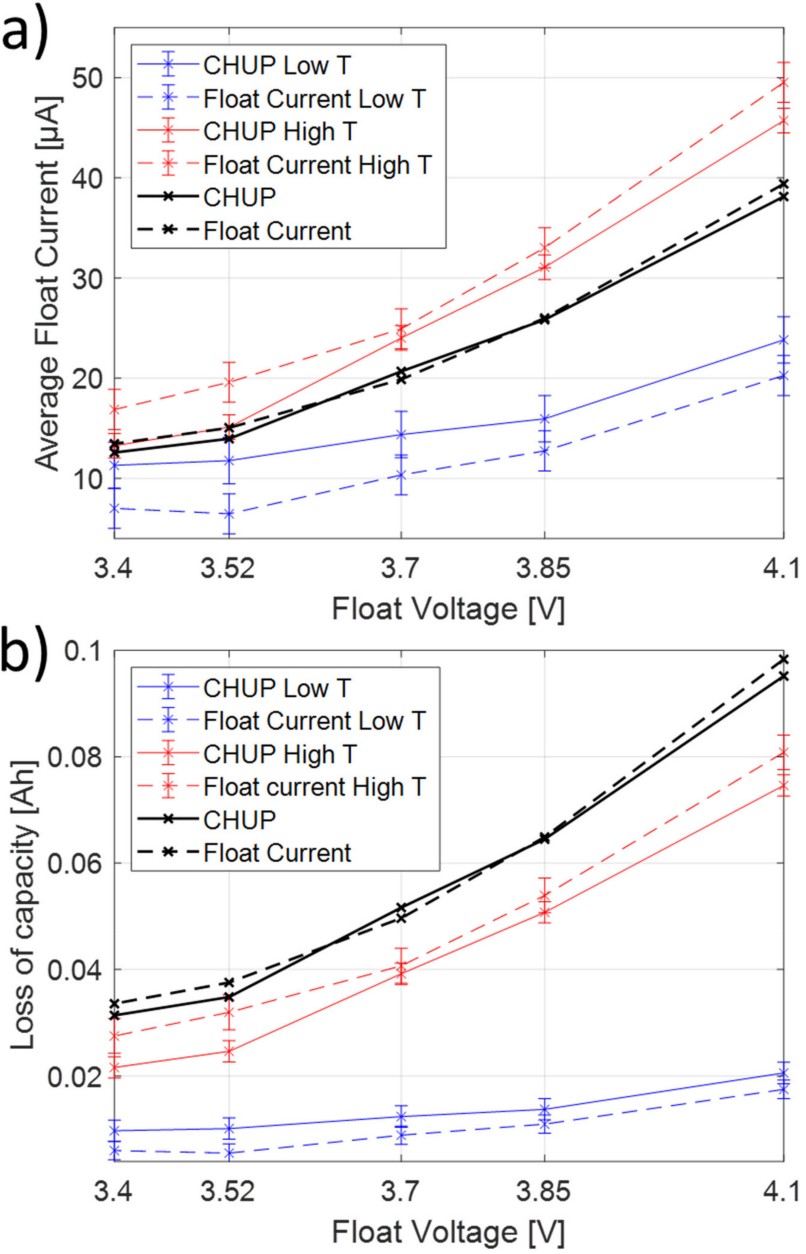

**Figure 13.** Capacity loss obtained from capacity tests (solid) and from float tests (dashed) over the float voltage of the cells for the low temperature cycle (blue), high temperature cycle (red) and both cycles combined (black) presented as: (**a**) average float current; and (**b**) loss of capacity.

There is a high correlation for both temperature sequences between capacity test results and float currents, and all curves show increasing losses with cell potential. For the low temperature sequence, for all values, the float currents are lower than for the check-up; for high temperatures, this is vice versa. The difference in capacity loss of both methods is in the low temperature cycle, with 3–5 mAh, corresponding to 0.1–0.2%, rather small. In the high temperature cycle, the difference is, with 2–7 mAh (0.1–0.3%), slightly stronger, especially for low and high float voltages. Considering both temperature sequences (black curves), the differences between the capacity losses obtained by both methods become even lower.

Thus, it can be concluded that the observed trends show a high correlation and that the absolute values are comparable considering the estimated error and that the offset is opposite for low and high temperature sequence. The lifetime can be predicted by integration of the float current until, e.g., 80% remaining capacity is reached. In this case, anode overhang effect is neglected. Repeating both temperature cycles continuously, the predicted lifetime ranges from 4.4 years at 3.4 V (13 µA) to 1.5 years at 4.1 V (38.5 µA).

The key question is still to name the root cause for the float current and why it is linked to capacity fade as shown before. The DVA results show mainly LLI losses and for the high temperature cycle additional cathode material losses that increase with cell potential. However, in Figure 13, we observe a difference between low and high temperature cycles by an offset but without dependence of cell potential. As the measured float current is higher than the capacity loss, we suggest that the cathode aging leads to a higher float current but not to a capacity loss in the same way as visible in Figure 13 by the offset. The main contribution to float current is assumed to originate from SEI formation during aging. How SEI formation triggers a self-discharge reaction that is recharged by a float current will be the scope of future publications.

### 3.6. Reproducibility of Float Currents at 30 °C

Finally, we check the reproducibility of the float current by path dependence at the reference temperature of 30 °C (see Figure 14). The reference value at 30 °C is measured before and after the low temperature sequence and after each temperature increase within the high temperature sequence. In the following, all these measurements are compared to each other.

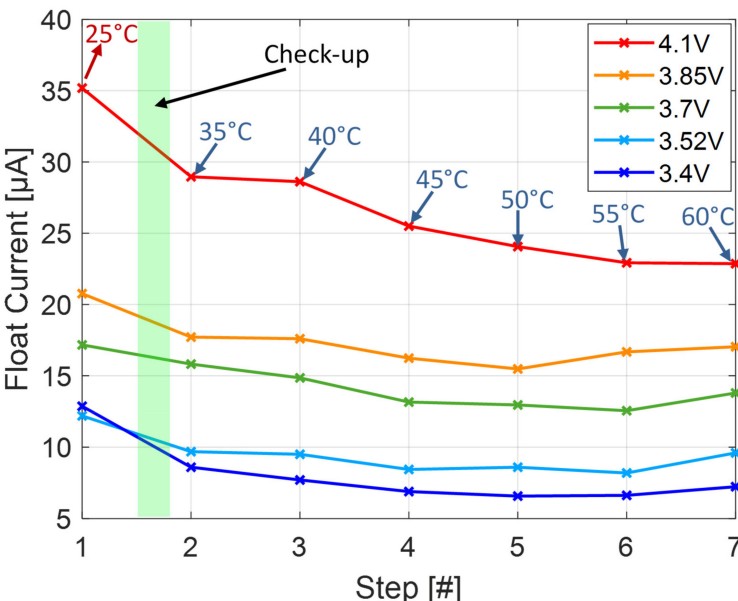

**Figure 14.** The float currents measured at reference temperature over the measurement steps.

The float current is decreasing with aging and/or increasing temperature drop. The drop is higher for higher float voltage. This can be caused by a hysteresis effect according

to temperature step and sign, as reported by Zilberman [33], and/or by reducing active surface due to aging at higher temperatures.

A decreasing active surface would reduce the float current in the same percentage, leading to significant deviations in the Arrhenius representation, which is not the observable towards higher temperatures. A constant offset of 10 µA for 4.1 V and 2–5 µA for all other float potentials would be hardly visible for higher temperatures in Figure 7 due to the already high float currents. As the results in Figure 7 were obtained after raising the temperature, the hysteresis theory is favored by the authors. However, more tests have to be conducted in the future to understand this phenomenon with more distinction.

## 4. Conclusions

The paper presents a novel method to assess the calendar aging without influences of anode overhang, resistance increase and disturbing reference condition during a check-up. It was shown that this method enables precise aging determination especially at low temperatures down to 10 °C in a reasonable time. Further, we demonstrated a fast characterization of calendar aging by evaluating float currents for five fixed cell potentials varying the temperature from 5 to 60 °C. As expected, the float current increased with higher cell potential and temperature and followed the Arrhenius law, and we obtained a typical activation energy as it is reported for capacity loss. The correlation between capacity loss from capacity tests and the float current was found qualitatively and quantitatively for low and high temperature cycle in very good agreement with only small deviations. The clear increase of resistance for 4.1 V showed no significant changes in float current at reference temperature, supporting the independence of resistance. The origin of the float current is associated mainly to SEI formation. There are some indications that loss of cathode material leads to a smaller additional float current increase compared to the measured capacity loss in this test. Finally, we found a path dependence of the float currents with respect to gradient of temperature changes towards 30 °C reference temperature resulting in a decreasing float current. A hysteresis phenomenon is expected to be the origin, but further experiments including temperature cycles will be needed.

**Author Contributions:** Conceptualization, M.L. and C.E.; methodology, M.L.; validation M.L. and M.T.; formal analysis, M.L. and M.T.; investigation, M.L. and M.T.; resources, M.L. and C.E.; data curation, M.T.; writing—original draft preparation, M.T. and M.L.; writing—review and editing, M.L. and C.E.; visualization, M.T.; supervision, C.E. and M.L.; project administration M.L.; and funding acquisition, M.L. All authors have read and agreed to the published version of the manuscript.

**Funding:** This research was funded by Deutsche Forschungsgemeinschaft (German Research Foundation, DFG) in the project "Research on fast and high-precision float current technology as an alternative to conventional calendar ageing tests on lithium ion cells and research on fast parameterizable ageing prognosis simulations", grant number LE 4469/1.

**Institutional Review Board Statement:** Not applicable.

**Informed Consent Statement:** Not applicable.

**Data Availability Statement:** The data presented in this study are openly available in FigShare at https://doi.org/10.6084/m9.figshare.14192390.v1 (accessed on 1 March 2020)

**Acknowledgments:** Special thanks are given to the support by Lidiya Komsiyska and Sascha Speer.

**Conflicts of Interest:** The authors declare no conflict of interest.

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
