# Peer review of "Float Current Analysis for Fast Calendar Aging Assessment of 18650 Li(NiCoAl)O2/Graphite Cells"

_batteries, doi:10.3390/batteries7020022_

Round 1

Reviewer 1 Report

The paper analysis the float current to assess the calendar aging of lithium-ion batteries. The authors use a test system from ARBIN in their experiments.

The manuscript presents a clear structure and easy to follow way. The paper topic is interesting and presentation of the simulated and experimental results are suitable.

However, the manuscript could be improved accordingly by clarifying the following:

  • What are the processes that lead to a degradation of a battery cell independent of charge-discharge cycling?
  • How does the capacity loss influence the remaining useful life of lithium-ion batteries?
  • What is the remaining useful life prediction in your experimental study?

Additional remarks:

  • Line 1: please delete "Type of the Paper"
  • Figure 14 was not cited in the main text.

Author Response

The paper analysis the float current to assess the calendar aging of lithium-ion batteries. The authors use a test system from ARBIN in their experiments.

The manuscript presents a clear structure and easy to follow way. The paper topic is interesting and presentation of the simulated and experimental results are suitable.

Thank you very much.

However, the manuscript could be improved accordingly by clarifying the following:

  • What are the processes that lead to a degradation of a battery cell independent of charge-discharge cycling?
  • How does the capacity loss influence the remaining useful life of lithium-ion batteries?

The degradation is given in lines 213-215

“In the low temperature sequence, only SEI formation is observed while in the high temperature sequence the cathode degrades significantly, too.”

We added for clarification here:

“Only LLI losses such as SEI formation are directly associated to loss of extractable capacity. Moderate losses of anode and cathode active materials do not necessarily lead to capacity fade as both electrodes are initially oversized in capacity and as this oversize is further increased by LLI loss over aging. However, if the deactivated active material encapsulates active lithium, this will lead to capacity fade but will appear as LLI losses and can therefore not be separated from SEI formation on the anode. Thus, the contribution of loss of cathode active material on capacity fade is limited and not fully determinable by DVA.”

Thus, we observe SEI formation as it is reported in many studies (10.1016/j.jpowsour.2005.01.006.). Additionally we observed loss of cathode capacity. We assume that SEI formation is associated to float current in lines 63-69

“Assuming that capacity loss is caused by solid electrolyte interphase (SEI) formation [20], as it is typically the case, intercalated active lithium-ions from the graphite are passivated in the SEI. As the slope of the open circuit potentials of the graphite potential is hardly zero, the existence of float currents cannot be explained directly by passivating lithium on the anode. A potential explanation is given in the publication by a parallel self-discharging shuttle reaction between the anode and the cathode triggered by SEI formation.”

In section 3.5 we added a link between capacity loss, DVA and float currents

“The key question is still to name the root cause for the float current and why it is linked to capacity fade as shown before. The DVA results showed mainly LLI losses and for the high temperature cycle additional cathode material losses that increase with cell potential. However in Fig. 13, we observe a difference between low and high temperature cycle by an offset but without dependence of cell potential. As the measured float current is higher than the capacity loss, we suggest that the cathode aging leads to a higher float current but not to a capacity loss in the same way as is visible in Fig. 13 by the offset. The main contribution to float current is assumed to originate from SEI formation during aging. How SEI formation triggers a self-discharge reaction that is recharged by a float current will be the scope of future publications.”

Finally we added to conclusion

“The origin of the float current is associated mainly to SEI formation. There are some indications that loss of cathode material leads to a smaller additional float current increase compared to the measured capacity loss in this test.”

  • What is the remaining useful life prediction in your experimental study?

We added a sentence giving an estimation of the useful life at 30°C.

“The lifetime can be predicted by integration of the float current until e.g. 80% remaining capacity is reached. In this case, anode overhang effect is neglected. Repeating both temperature cycles continuously, the predicted lifetime ranges from 4.4 years at 3.4 V (13 µA) to 1.5 years at 4.1 V (38.5 µA).”

Additional remarks:

  • Line 1: please delete "Type of the Paper"
  • Figure 14 was not cited in the main text.

Thank you very much. We corrected the reviewer’s statements.

Reviewer 2 Report

The present manuscript reports a calendar aging assessment method based on float current analysis. The manuscript is clear and sound and well structured. Overall, hypotheses and conclusions are supported by evidence and data.

After careful reading, this reviewer would like make a few comments which are listed below:

  • In lines 92-93, the last sentence of the paragraph stands "Finally, in section the hysteresis...". Perhaps section number is missing.
  • Section 2.1 describes the check-up procedures. This paragraph, although giving precise information, could be somewhat difficult to follow. Perhaps, adding a table collecting the information about charge/discharge procedure of the check-up could be useful to the readers.
  • Table 1 describes the battery specifications. The cathodes, a mixture of NCA and NCM materials as it seems, are displayed as Li(NiCoAl)O2 + Li(Ni6Mn2Co2)O2. This may be not correct, since the amount of transition metals in both layered oxides should sum 1, having the general formula LiMO2. Therefore, Li(NixCoyAlz)O2 (when proportions are unknown) would be more appropriate. On the other hand, the NCM material should be Li(Ni0.6Mn0.2Co0.2)O2 (the so called NCM 622) also with the sum of T.M. equal to 1.
  • Also, in line 321 the NCM material should show the formula given in the last point.
  • Figure 5a has the legend over the curves. Could it be possible to move the legend?
  • Figure caption of Fig.13 assigns blue and red curves to a different concept than in the text (lines 334-346). Should it mean solid and dashed lines instead of red and blue? Please check.

In summary, this reviewer finds the present manuscript very interesting and suitable for publication after the minor revision proposed. 

Author Response

The present manuscript reports a calendar aging assessment method based on float current analysis. The manuscript is clear and sound and well structured. Overall, hypotheses and conclusions are supported by evidence and data.

Thank you very much.

After careful reading, this reviewer would like make a few comments which are listed below:

  • In lines 92-93, the last sentence of the paragraph stands "Finally, in section the hysteresis...". Perhaps section number is missing.

Thank you. The section number was added.

  • Section 2.1 describes the check-up procedures. This paragraph, although giving precise information, could be somewhat difficult to follow. Perhaps, adding a table collecting the information about charge/discharge procedure of the check-up could be useful to the readers.

We added table 2 for a summary.

Table 2 Check-up test

Step

Mode

Values

Pause

1

CCCV charge

at 1C to 4.2V until I < 0.25A

15 min

2

CC discharge

at 1C to 2.5V

15 min

3

CCCV charge

at 1C to 4.2V until I < 0.25A

15 min

4

CC discharge

at 0.1C to 2.5V

15 min

5

CC charge

at 0.1C to 4.2V

15 min

6

CC discharge

at 0.1C to 3.7V

15 min

7

Pulse test

1C discharge; 10 s

8

CC discharge

at 0.1C to 2.5V

9

CC charge

at 0.1C to float voltage

  • Table 1 describes the battery specifications. The cathodes, a mixture of NCA and NCM materials as it seems, are displayed as Li(NiCoAl)O2 + Li(Ni6Mn2Co2)O2. This may be not correct, since the amount of transition metals in both layered oxides should sum 1, having the general formula LiMO2. Therefore, Li(NixCoyAlz)O2 (when proportions are unknown) would be more appropriate. On the other hand, the NCM material should be Li(Ni0.6Mn0.2Co0.2)O2 (the so called NCM 622) also with the sum of T.M. equal to 1.

Thank you for highlighting this. We changed it accordingly.

  • Also, in line 321 the NCM material should show the formula given in the last point.

changed

  • Figure 5a has the legend over the curves. Could it be possible to move the legend?

We changed the alignment to show all curves.

  • Figure caption of Fig.13 assigns blue and red curves to a different concept than in the text (lines 334-346). Should it mean solid and dashed lines instead of red and blue? Please check.

 Thank you. We corrected the error originated from earlier manuscript versions.

In summary, this reviewer finds the present manuscript very interesting and suitable for publication after the minor revision proposed. 

Thank you for your valuable comments

Reviewer 3 Report

Authors reported a study about the aging of commercial LIBs.

This work lacks of relevance from a scientific point of view. What is the advancement of knowledge promoting by this study? In the best case scenario this is represent only an advanced technical report usefully for company internal use. 

Furthermore,  data are reported without any statistical analysis and associated uncertains.

I strongly discourage the pubblication of this paper because it dos not meet the minimum requirements for pubblication in a peer review reputated journal.

Author Response

  • Authors reported a study about the aging of commercial LIBs.

Dear reviewer, we like to thank you for taking time to review the manuscript.

  • This work lacks of relevance from a scientific point of view. What is the advancement of knowledge promoting by this study? In the best case scenario this is represent only an advanced technical report usefully for company internal use.

We are astonished and actually rather confused by the comments of the reviewer.

In our opinion to develop faster methodology to assess and quantify the aging of battery cells  is of major relevance not only to the industry, but also to the scientific community. Quantification of the cell aging is performed when new material are developed, cells are produced and updated, battery systems are designed and lifetime prediction models are parametrized. The advantage of this method is that

  • we are not measuring anode overhang effect and with this achieve a higher precision (10.1149/2.0191503jes.)
  • we are able to measure aging even at low temperatures
  • we can assess capacity losses over temperature in small steps (with check-ups we are limited to mostly 3 temperatures due to costs and limited infrastructure)

Thus, the lifetime prediction can be improved and cells can be characterized faster and with a higher precision. Possible deviations of the correlation are of major interest and may help understanding in more depth the aging mechanisms.

  • Furthermore,  data are reported without any statistical analysis and associated uncertains.

We have to admit that we have no statistical measurements as only five cells have been tested due to limitation in test channels. This will be done in future work with several cells in the same conditions. Recent results for another cells show a very high reproducibility. This fits to typical test results for calendar aging results, especially for lower temperatures and voltages, which have a high reproducibility (10.1016/j.est.2018.04.029, 10.1016/j.est.2020.101547).

The goal with statistics is to support the data’s reproducibility and the reliability of the experiment. However, there are also other ways to come to this point:

  • We investigated 5 cell potentials that show a clear Arrhenius behaviour
  • We compared the float currents to capacity loss with a high correlation by pattern and by absolute value
  • The activation energy fits perfectly to capacity loss reported in literature
  • The 3-D aging pattern fits very well to reported aging representations in literature.

Comparable approach is done by Keil et al. (10.1149/2.0411609jes) measuring one cell for every 5% SOC instead of testing several cells at one condition.

  • I strongly discourage the pubblication of this paper because it dos not meet the minimum requirements for pubblication in a peer review reputated journal.

We hope that the additional explanations we provided changed your opinion on the merit and significance of our contribution.

Round 2

Reviewer 3 Report

Please consider that methodology and scientific approach herein described is not  up to mark of the most advance research in the field (Anselma PG, Kollmeyer P, Lempert J, Zhao Z, Belingardi G, Emadi A. Battery state-of-health sensitive energy management of hybrid electric vehicles: Lifetime prediction and ageing experimental validation. Applied Energy. 2021 Mar 1;285:116440; Hosen MS, Jaguemont J, Van Mierlo J, Berecibar M. Battery lifetime prediction and performance assessment of different modeling approaches. Iscience. 2021 Jan 19:102060; Atalay, S., Sheikh, M., Mariani, A., Merla, Y., Bower, E. and Widanage, W.D., 2020. Theory of battery ageing in a lithium-ion battery: Capacity fade, nonlinear ageing and lifetime prediction. Journal of Power Sources478, p.229026.). I appreciate the tentative of temperature related study but the assuming that the limited infrastructures could be adressed for the quality of a research is missleading  and only results are the measurment of the quality for a scientific work

Furtheremore as reported by Attia et al.(Attia PM, Severson KA, Witmer JD. Statistical learning for accurate and interpretable battery lifetime prediction. arXiv preprint arXiv:2101.01885. 2021 Jan.), a strong statisticall approach is more than a merely requirement. Additionally, a single data without any repetition and related error is not considerable as reproducible. 

Authors did not solve any issue of the paper  and just added few text lines.

Author Response

Reviewer 3

Please consider that methodology and scientific approach herein described is not  up to mark of the most advance research in the field

 (Anselma PG, Kollmeyer P, Lempert J, Zhao Z, Belingardi G, Emadi A. Battery state-of-health sensitive energy management of hybrid electric vehicles: Lifetime prediction and ageing experimental validation. Applied Energy. 2021 Mar 1;285:116440;

Hosen MS, Jaguemont J, Van Mierlo J, Berecibar M. Battery lifetime prediction and performance assessment of different modeling approaches. Iscience. 2021 Jan 19:102060;

Atalay, S., Sheikh, M., Mariani, A., Merla, Y., Bower, E. and Widanage, W.D., 2020. Theory of battery ageing in a lithium-ion battery: Capacity fade, nonlinear ageing and lifetime prediction. Journal of Power Sources478, p.229026.).

I appreciate the tentative of temperature related study but the assuming that the limited infrastructures could be adressed for the quality of a research is missleading  and only results are the measurment of the quality for a scientific work

Furtheremore as reported by Attia et al.

(Attia PM, Severson KA, Witmer JD. Statistical learning for accurate and interpretable battery lifetime prediction. arXiv preprint arXiv:2101.01885. 2021 Jan.)

, a strong statisticall approach is more than a merely requirement. Additionally, a single data without any repetition and related error is not considerable as reproducible. 

Authors did not solve any issue of the paper  and just added few text lines.

Dear Reviewer, we agree to your statements and to the named papers and the significance of that work. However, you cited here papers that investigated the following

  • Cycle life
  • Aging until rollover

For these conditions, you are perfectly right. In this manuscript, we have pure calendar aging at low aging conditions. No rollover at all. For these measurements, it is less important to check for statistical issues due to the low cell-to-cell variation as we cited in our previous reply. Thus, you have to admit: This is a difference.

To sum up: in this test we make prognosis of aging for very mild aging conditions. The higher temperatures have been applied only for a very short time. Thus you can conclude that the aging is far away from rollover. The results for the correlation to capacity fade at low temperature and high temperature cycle are comparable so that we do not have any hint for stronger aging at higher temperatures.

We respect that you come from statistical background. However, you can make points as well by changing parameters that follow e.g. the Arrhenius behaviour or show a high correlation to capacity fade for each of the five cells. We will consider your comments for future publications but until now, we are more focused on looking a variety of cells with different active materials to see if we have here a correlation and at which point there is no correlation.  

To highlight your issues we added to introduction part:

“In this contribution, we present a fast parametrization of the calendar aging rate as a function of the cell potential and the temperature for five 18650 NCA/graphite cells. The resolution at low temperatures is highlighted between 0-10°C. Finally, the path dependence of float currents is discussed. The data is correlated for each cell separately by comparing capacity losses rate and float current. The aging and the float current are further discussed by the active material losses obtained from differential voltage analysis. Moreover, the trend temperature dependence over SOC for all five cells is evaluated by means of the Arrhenius law. The cell-to-cell variation during calendar aging is beyond the scope of this publication and is expected to be comparatively low, especially for lower SOCs as it is shown in many publications before [3,5,9,27]. The test strategy including a low temperature and a high temperature sequence is presented in chapter 2. Then at first, the check-ups are evaluated with standard methods in section 3.1, followed by the float current analysis in section 3.2-3.4. In section 3.5, results of both methods are compared with each other. Finally, in section 3.6 the hysteresis and reproducibility at 30°C is discussed.”